# Clostridioides (Clostridium) difficile infection in hospitalized adult patients in Cambodia

Lengsea Eng,[1,2] Deirdre A. Collins,[1,3,4] Kefyalew Addis Alene,[1,5] Sotharith Bory,[2,6] Youdaline Theng,[2,6] Pisey Vann,[2] Sreyhuoch Meng,[2,6] Setha Limsreng,[2,6] Archie C. A. Clements,[1] Thomas V. Riley[1,4,7]

**ABSTRACT**  Despite high levels of global concern, little is known about the epidemiology of Clostridioides (Clostridium) difficile infection (CDI) in Cambodia. This study aimed to identify the prevalence and risk factors for CDI, and molecular types of C. difficile in hospitalized adults at Calmette Hospital, Phnom Penh, Cambodia. Stool samples were collected from 263 hospitalized adults between June and September 2022 and tested for C. difficile using direct and enrichment cultures. PCR toxin genes tcdA, tcdB, cdtA, and cdtB, and amplification of the 16s–23s rRNA intergenic spacer region for ribotyping, were performed on all C. difficile isolates. C. difficile was isolated from 24% (63/263) of samples, and most isolates were non-toxigenic (67%, 42/63). The five most predominant toxigenic C. difficile ribotypes (RTs) were RTs 046 (8%, 5/63), 017 (6%, 4/63), 056 (5%, 3/63), 014/020 (5%, 3/63), and 012 (3%, 2/63), and prominent non-toxigenic RTs were QX011 (14%, 9/63), 010 (8%, 5/63), 009 (3%, 2/63), QX021 (3%, 2/63), and QX002 (3%, 2/63). Risk factors significantly associated with CDI included diabetes (odds ratio [OR] = 2.48, 95% confidence interval [CI]: 1.16–5.30) and hospitalization >24 h within the last 3 months before testing (OR = 3.89, 95% CI: 1.79–8.43). It was concluded that most participants from whom C. difficile was isolated were colonized only; however, a high prevalence of asymptomatic carriage could contribute to silent transmission in healthcare settings and communities. Genotypic identification of local C. difficile strains is necessary for a better understanding of the epidemiology of CDI and the importance of C. difficile.

**IMPORTANCE**  Clostridioides difficile is a significant cause of diarrhea worldwide, initially as a hospital-acquired infection and, more recently, as a community-associated infection. Risk factors for hospital-acquired C. difficile infection include antimicrobial consumption, extended hospitalization, age ≥ 65 years, and proton pump inhibitor treatment. While much is known about C. difficile in high-income countries, little is known and there has been less interest in this infection in Asia due to the lack of data. Thus, investigating the prevalence and risk factors for C. difficile and characterizing C. difficile strains from hospitalized adults is necessary in Asian countries such as Cambodia. Diabetes and hospitalization >24 h within the last 3 months were identified as risk factors for C. difficile colonization/infection. The high prevalence of non-toxigenic strains and asymptomatic carriage of C. difficile in this country were notable. Further studies are warranted to gain better insights into this infection in Cambodia.

**KEYWORDS**  Clostridioides difficile, Cambodia, risk factors, ribotype, healthcare

Clostridioides (Clostridium) difficile infection (CDI) is the most reported hospital-acquired infection in high-income countries worldwide with considerable challenges in patient management (1–3). Symptoms range from mild to severe diarrhea through toxic megacolon and pseudomembranous colitis with a high proportion of asymptomatic infection. The US Centers for Disease Control and Prevention ranked C.

**Peer Reviewer** Dazhi Jin, Hangzhou Medical College, Hangzhou, China

Address correspondence to Lengsea Eng, 20018850@student.curtin.edu.au, lengsea.eng@postgrad.curtin.edu.au, or Thomas V. Riley, thomas.riley@uwa.edu.au.

The authors declare no conflict of interest.

*difficile* as an urgent antimicrobial resistance (AMR) threat in the USA, costing the US healthcare system ~USD 1 billion annually (3). Individuals with CDI and, particularly, recurrent CDI have longer stays in hospitals and are at significantly higher risk of complications and mortality compared to those without CDI (4). Historical outbreaks of CDI in North America and Europe raised concerns about *C. difficile* transmission and changes in epidemiology across the globe (5–10). An extensive list of CDI risk factors includes antimicrobial therapy, aging, proton pump inhibitor (PPI) treatment, extended hospitalization, carriage of multidrug-resistant organisms, comorbidities such as diabetes, renal diseases, and heart failure, and low socio-economic status (11–15).

Several studies have demonstrated that, in Asia, most infections with *C. difficile* are asymptomatic (16–18). Many such infections are not identified due to the subclinical nature of the disease presentation, lack of awareness of CDI clinical diagnosis, or the unavailability of diagnostic facilities (16, 19, 20). The prevalence of CDI in Asia has varied considerably in studies, ranging from 7% to 37.8% (16, 19, 21–26). A recent multi-center study of CDI in countries in the Asia-Pacific region including Indonesia, Malaysia, Singapore, Thailand, Vietnam, The Philippines, and Australia revealed that the predominant circulating *C. difficile* strain in Asia was ribotype (RT) 017 followed by RTs 002, 018/QX239, and 014/020 with a wide range of genotypes of *C. difficile* across the studied countries (11). Cambodia was not included in this study, and there are only two early publications on *C. difficile* in Cambodia; a study of *C. difficile* in HIV-infected patients (prevalence 3.75%) (27) and one about *C. difficile* in dried and smoked fish (16%) (28).

While the cost of CDI management is significant and morbidity can be high, Cambodia lacks any epidemiological data on CDI to aid in the management of clinical and public health interventions. Thus, this study aimed to investigate the prevalence, risk factors, and molecular epidemiology of CDI in hospitalized adult patients in Cambodia.

## RESULTS

### Characteristics

A total of 263 patients were included during the 4-month sample collection period, with 12 patients hospitalized for <48 h at the time of sample collection. To evaluate potential differences in risk factors for *C. difficile* carriage that had been acquired in the hospital, separate analyses were conducted for all 263 patients and for the subset of 251 patients who had been hospitalized for ≥48 h.

Female patients represented 55% (143/262) of the cohort, while 34% (89/261) were aged ≥65 years. A total of 63 patients were positive for *C. difficile* carriage (24%), identified through both direct and enrichment cultures. Among the 251 patients hospitalized for ≥48 h, the prevalence of *C. difficile* carriage was 23.1% (58/251), with no significant difference in prevalence compared to the complete cohort ($P = 0.821$). In patients presenting with diarrhea, the prevalence of *C. difficile* was 15.2% (5/33) (Fig. 1). The prevalence among the 12 patients hospitalized for <48 h was 41.7% (5/12), which was not significantly higher than in the entire cohort ($P = 0.164$) or the subgroup of 251 patients ($P = 0.141$). Notably, none of the 12 patients hospitalized for <48 h had diarrhea.

Patients were residents of 19 different regions of Cambodia, and *C. difficile* was detected in patients from 16 of those regions (Fig. 2). In regions with more than 10 participants, except for Kampong Thom, the prevalence of *C. difficile* ranged from 13% to 67%. The five patients with *C. difficile* hospitalized <48 h were not residents of Phnom Penh and were not hospitalized in other hospitals before Calmette Hospital (CH).

### Risk factors

In the univariate analysis of the 263 patients, several demographic and clinical factors were associated with *C. difficile* carriage (Table 1). These included diabetes (odds ratio [OR] = 1.88; 95% confidence interval [CI]: 1.06–3.36), frequent outpatient department (OPD) visits (OR = 2.44; 95% CI: 1.13–5.27), treatment with PPIs (OR = 8.06; 95% CI: 1.07–60.98), hemodialysis (OR = 6.64; 95% CI: 1.19–37.18), hospitalization >24 h within the last

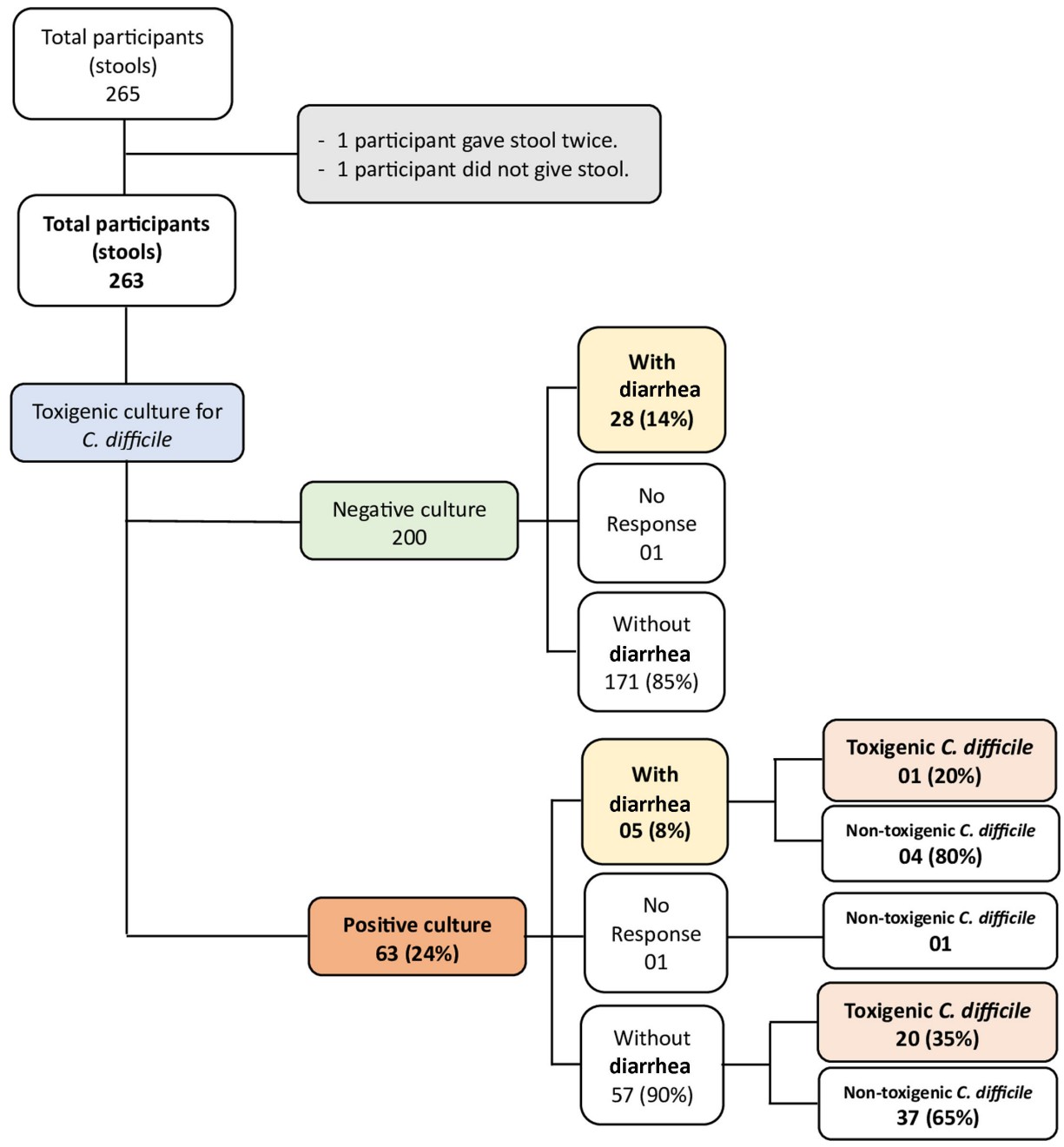

**FIG 1** Study participants and *C. difficile* toxigenic culture findings, by diarrheic status.

3 months (OR = 2.17; 95% CI: 1.19–3.98), residing outside Phnom Penh (OR = 2.39; 95% CI: 1.02–5.64), and abdominal pain (OR = 1.89; 95% CI: 1.02–3.50) (Table 1). There was no association between conventional risk factors such as age ≥ 65 years, length of stay (LOS) in hospitals, antimicrobial consumption, and *C. difficile* carriage (Table 1). When the analysis was repeated for the subset of 251 patients hospitalized for ≥48 h, the results were largely consistent with the 263-patient group. However, residing outside Phnom Penh was no longer significantly associated with *C. difficile* carriage in this subgroup, whereas it reached statistical significance in the full 263-patient analysis ($P = 0.046$).

Multivariable analysis demonstrated that diabetes (OR = 2.48; 95% CI: 1.16–5.30) and hospitalization >24 h within the last 3 months (OR = 3.89; 95% CI: 1.79–8.43) were significantly associated with the presence of *C. difficile* (Table 2). Nephrotic syndrome and esophageal variceal bleeding (OVB), despite having a high proportion of positive *C.*

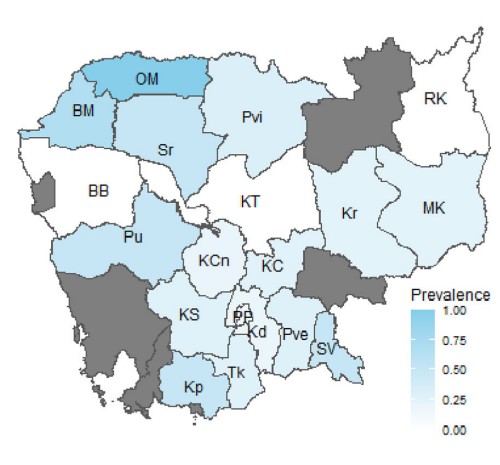

| Region | Number of participants | Number with *C. difficile* | RT046 | RT017 | RT056 | RT 014/020 | RT012 | Others |
|---|---|---|---|---|---|---|---|---|
| 1. BM, Banteay Meanchey | 3 | 2 (67%) | | | | | | 2 |
| 2. BB, Battambang | 3 | 0 | | | | | | |
| 3. KC, Kampong Cham | 30 | 8 (27%) | | 1 | | 1 | | 6 |
| 4. KCn, Kampong Chhnang | 10 | 2 (20%) | | 1 | | | 1 | |
| 5. KS, Kampong Speu | 21 | 6 29%) | 2 | 1 | 1 | | | 2 |
| 6. KT, Kampong Thom | 12 | 0 | | | | | | |
| 7. Kp, Kampot | 6 | 3 (50%) | | | | | | 3 |
| 8. Kd, Kandal | 44 | 8 (18%) | | 1 | | | 1 | 6 |
| 9. Kr, Kratie | 4 | 1 (25%) | | | | | | 1 |
| 10. MK, Mondul Kiri | 4 | 1 (25%) | | | | | | 1 |
| 11. OM, Oddar Meanchey | 1 | 1 (100%) | | | | | | 1 |
| 12. PP, Phnom Penh | 53 | 7 (13%) | 3 | | | | | 4 |
| 13. Pvi, Preah Vihear | 3 | 1 (33%) | | | | | | 1 |
| 14. Pve, Prey Veng | 24 | 7 (29%) | | | 1 | 1 | | 5 |
| 15. Pu, Pursat | 4 | 2 (50%) | | | | | | 2 |
| 16. RK, Ratanak Kiri | 2 | 0 | | | | | | |
| 17. Sr, Siemreap | 4 | 2 (50%) | | | | | | 2 |
| 18. SV, Svay Rieng | 6 | 3 (50%) | | | 1 | | | 2 |
| 19. Tk, Takeo | 14 | 4 (29%) | | | | 1 | | 3 |
| Total | 248 | 248 (100%) | | | | | | |

**Note:** Regions in gray had no cases.

**FIG 2** Number of participants and prevalence of *C. difficile* carriage, by region.

*difficile* cases, were not included in the logistic regression model due to the small sample size (Table S1). Notably, two out of three patients with nephrotic syndrome and three out of four patients with OVB had *C. difficile* cultured from their stool.

Among the 63 patients from whom *C. difficile* was isolated, nine and two of them were receiving metronidazole and vancomycin treatment, respectively, and one patient was receiving both at the time of stool collection (Table S2). One of the positive cases was on vancomycin therapy, 1 week and 4 weeks before sample collection, and another was on metronidazole treatment the week before sample collection. These antimicrobials were given for non-*C. difficile*-related issues. Antimicrobial consumption in the week prior to sample collection was seen in 42% (25/59) and in the 4 weeks prior to sample collection in 9% (5/56) of *C. difficile*-positive patients.

Only five of the total *C. difficile*-positive patients had diarrhea (defined as ≥3 loose stools within 24 h). Features of these five patients are shown in Table 3. Most of these patients (at least 3/5) had diabetes and renal diseases and were taking corticosteroids, PPIs, and other medications, and had antimicrobials, PPIs, and other medications in the week prior to stool collection.

## Ribotyping and toxin profiling

*C. difficile* isolates were identified as belonging to nine known RTs; however, 27 isolates could not be given an RT based on comparing banding patterns to an extensive collection of reference strains in the laboratory and were thus given a QX prefix. Toxigenic strains represented 33% (21/63) of isolates, of which *C. difficile* RT 046 (8%, 5/63) was predominant, followed by RTs 017 (6%, 4/63), 056 (5%, 3/63), 014/020 (5%, 3/63), and 012 (3%, 2/63) (Fig. 3). Most toxigenic strains (76%, 16/21), including unidentified strains, contained both toxin A and B genes. Non-toxigenic *C. difficile* represented 67% (42/63) of isolates, the most predominant being *C. difficile* RT QX011 (14%, 9/63) followed by RTs 010 (8%, 5/63), 009 (3%, 2/63), QX021 (3%, 2/63), and QX002 (3%, 2/63) (Fig. 3). Besides *C. difficile* RTs 017, 046, 056, 009, 010 and QX011, other RTs were not shared between the two wards, General Medicine Ward A (MA) and General Medicine Ward B (MB) (Fig. 3). Toxigenic *C. difficile* strains were found mostly in patients from provinces surrounding the capital, Phnom Penh (Fig. 2).

TABLE 1 Univariate logistic regression analysis of socio-demographic, clinical, and other factors associated with *C. difficile* carriage[a,b]

| | Number of cases | | COR (95% CI) | *P* value | COR (95% CI) | *P* value |
|---|---|---|---|---|---|---|
| | Without *C. difficile* | With *C. difficile* | (263 patients) | | (251 patients) | |
| | (*N* = 200) | (*N* = 63) | | | | |
| Socio-demographic factor | | | | | | |
| Sex (female) | | | | | | |
| Yes | 111 | 32 | 1.17 (0.66–2.07) | 0.591 | 1.10 (0.61–1.98) | 0.760 |
| No | 89 | 30 | 1.00 | | 1.00 | |
| Age (IQR: 44–68) | | | | | | |
| ≥65 years | 64 | 25 | 1.42 (0.79–2.56) | 0.238 | 1.45 (0.79–2.67) | 0.230 |
| <65 years | 135 | 37 | 1.00 | | 1.00 | |
| Residency: living outside the capital city | | | | | | |
| Yes | 143 | 51 | **2.39 (1.02–5.64)** | **0.046** | 2.14 (0.90–5.08) | 0.084 |
| No | 47 | 7 | 1.00 | | 1.00 | |
| Living close to livestock | | | | | | |
| Yes | 56 | 14 | 0.72 (0.37–1.42) | 0.346 | 0.61 (0.29–1.27) | 0.183 |
| No | 139 | 48 | 1.00 | | 1.00 | |
| Live with siblings <1 year old | | | | | | |
| Yes | 7 | 2 | 0.92 (0.19–4.54) | 0.918 | 0.97 (0.19–4.79) | 0.966 |
| No | 193 | 60 | 1.00 | | 1.00 | |
| Feature | | | | | | |
| Length of stay (IQR: 3–8 days, median = 5 days) | Median = 5 | Median = 6 | 1.01 (0.95–1.06) | 0.834 | 1.11 (0.96–1.07) | 0.568 |
| Blood leucocyte > 9 Giga/L | | | | | | |
| Yes | 83 | 28 | 1.00 (0.55–1.81) | 0.998 | 0.80 (0.44–1.48) | 0.484 |
| No | 89 | 30 | 1.00 | | 1.00 | |
| Stool consistency (loose) | | | | | | |
| Yes | 19 | 7 | 0.74 (0.21–2.58) | 0.633 | 0.91 (0.25–3.31) | 0.883 |
| No | 178 | 56 | 1.00 | | 1.00 | |
| Diarrhea | | | | | | |
| Yes | 28 | 5 | 0.53 (0.19–1.44) | 0.216 | 0.56 (0.21–1.53) | 0.261 |
| No | 170 | 57 | 1.00 | | 1.00 | |
| Abdominal pain | | | | | | |
| Yes | 44 | 22 | **1.89 (1.02–3.50)** | **0.043** | **1.91 (1.01–3.59)** | **0.045** |
| No | 156 | 41 | 1.00 | | 1.00 | |
| Fever | | | | | | |
| Yes | 58 | 23 | 1.40 (0.77–2.54) | 0.272 | 1.34 (0.72–2.50) | 0.349 |
| No | 141 | 40 | 1.00 | | 1.00 | |
| Hepatitis/cirrhosis | | | | | | |
| Yes | 16 | 3 | 0.57 (0.16–2.04) | 0.392 | 0.60 (0.17–2.15) | 0.436 |
| No | 184 | 60 | 1.00 | | 1.00 | |
| Anaemia and other blood disorders | | | | | | |
| Yes | 15 | 2 | 0.40 (0.09–1.82) | 0.238 | 0.46 (0.10–2.07) | 0.309 |
| No | 185 | 61 | 1.00 | | 1.00 | |
| Pneumonia/COPD | | | | | | |
| Yes | 54 | 18 | 1.08 (0.58–2.03) | 0.807 | 1.03 (0.54–1.99) | 0.923 |
| No | 146 | 45 | 1.00 | | 1.00 | |
| Acute pneumonia oedema | | | | | | |
| Yes | 9 | 3 | 1.06 (0.28–4.04) | 0.931 | 1.70 (0.41–7.02) | 0.463 |
| No | 191 | 60 | 1.00 | | 1.00 | |
| Other respiratory/lung conditions | | | | | | |
| Yes | 23 | 2 | 0.25 (0.06–1.10) | 0.067 | 0.28 (0.06–1.22) | 0.089 |
| No | 177 | 61 | 1.00 | | 1.00 | |
| Urinary tract infection | | | | | | |

(*Continued on next page*)

TABLE 1 Univariate logistic regression analysis of socio-demographic, clinical, and other factors associated with *C. difficile* carriage[a,b] (*Continued*)

| | Number of cases | | COR (95% CI) | *P* value | COR (95% CI) | *P* value |
|---|---|---|---|---|---|---|
| | Without *C. difficile* | With *C. difficile* | (263 patients) | | (251 patients) | |
| | (*N* = 200) | (*N* = 63) | | | | |
| Yes | 17 | 3 | 0.54 (0.15–1.90) | 0.336 | 0.37 (0.08–1.65) | 0.192 |
| No | 193 | 60 | 1.00 | | 1.00 | |
| Other bacterial infections | | | | | | |
| Yes | 9 | 5 | 1.83 (0.59–5.68) | 0.296 | 1.93 (0.62–6.00) | 0.257 |
| No | 191 | 58 | 1.00 | | 1.00 | |
| Cholecystitis | | | | | | |
| Yes | 3 | 3 | 3.28 (0.65–16.69) | 0.152 | 3.45 (0.68–17.60) | 0.136 |
| No | 197 | 60 | 1.00 | | 1.00 | |
| Cardiopathy | | | | | | |
| Yes | 7 | 4 | 1.87 (0.53–6.61) | 0.332 | 1.45 (0.36–5.79) | 0.600 |
| No | 193 | 59 | 1.00 | | 1.00 | |
| Heart failure | | | | | | |
| Yes | 24 | 12 | 1.73 (0.81–3.69) | 0.159 | 1.59 (0.68–3.71) | 0.285 |
| No | 176 | 51 | 1.00 | | 1.00 | |
| Bowel function disorders | | | | | | |
| Yes | 2 | 3 | 4.95 (0.81–30.32) | 0.084 | 5.21 (0.85–31.96) | 0.075 |
| No | 198 | 60 | 1.00 | | 1.00 | |
| Diabetes | | | | | | |
| Yes | 65 | 30 | **1.89 (1.06–3.36)** | **0.031** | **1.88 (1.04–3.43)** | **0.038** |
| No | 135 | 33 | 1.00 | | 1.00 | |
| Cortico-adrenal insufficiency | | | | | | |
| Yes | 9 | 4 | 1.44 (0.43–4.84) | 0.557 | 1.97 (0.56–6.98) | 0.294 |
| No | 191 | 59 | 1.00 | | 1.00 | |
| Renal diseases | | | | | | |
| Yes | 33 | 16 | 1.72 (0.87–3.40) | 0.116 | 1.80 (0.88–3.70) | 0.110 |
| No | 167 | 47 | 1.00 | | 1.00 | |
| Hypertension | | | | | | |
| Yes | 67 | 28 | 1.59 (0.89–2.82) | 0.116 | 1.53 (0.84–2.78) | 0.167 |
| No | 133 | 35 | 1.00 | | 1.00 | |
| Stroke | | | | | | |
| Yes | 8 | 2 | 0.78 (0.16–3.78) | 0.761 | 0.40 (0.05–3.30) | 0.397 |
| No | 191 | 61 | 1.00 | | 1.00 | |
| Cancer | | | | | | |
| Yes | 12 | 4 | 1.05 (0.33–3.37) | 0.941 | 1.10 (0.34–3.55) | 0.875 |
| No | 185 | 59 | 1.00 | | 1.00 | |
| Frequent OPD visit (at least 1/week) | | | | | | |
| Yes | 19 | 13 | **2.44 (1.13–5.27)** | **0.024** | **2.49 (1.12–5.55)** | **0.025** |
| No | 178 | 50 | 1.00 | | 1.00 | |
| Non-surgical GI procedure | | | | | | |
| Yes | 40 | 16 | 1.35 (0.69–2.62) | 0.383 | 1.44 (0.73–2.82) | 0.290 |
| No | 158 | 47 | 1.00 | | 1.00 | |
| Hemodialysis | | | | | | |
| Yes | 2 | 4 | **6.64 (1.19–37.18)** | **0.031** | **7.00 (1.25–39.25)** | **0.027** |
| No | 196 | 59 | 1.00 | | 1.00 | |
| Surgery | | | | | | |
| Yes | 27 | 12 | 1.47 (0.70–3.11) | 0.311 | 1.57 (0.74–3.33) | 0.245 |
| No | 169 | 51 | 1.00 | | 1.00 | |
| 3 months prior to stool collection | | | | | | |
| Hospitalization >24 h | | | | | | |
| Yes | 46 | 25 | **2.17 (1.19–3.98)** | **0.012** | **2.96 (1.23–4.28)** | **0.009** |

(*Continued on next page*)

**TABLE 1** Univariate logistic regression analysis of socio-demographic, clinical, and other factors associated with *C. difficile* carriage[a,b] (*Continued*)

| | Number of cases | | COR (95% CI) | P value | COR (95% CI) | P value |
|---|---|---|---|---|---|---|
| | Without *C. difficile* (N = 200) | With *C. difficile* (N = 63) | (263 patients) | | (251 patients) | |
| No | 148 | 37 | 1.00 | | 1.00 | |
| Frequent OPD visits (at least once a week) | | | | | | |
| Yes | 15 | 9 | 2.07 (0.86–5.00) | 0.105 | 1.92 (0.77–4.78) | 0.164 |
| No | 183 | 53 | 1.00 | | 1.00 | |
| Reside in a long-term care facility | | | | | | |
| Yes | 8 | 3 | 1.19 (0.31–4.65) | 0.797 | 1.26 (0.32–4.90) | 0.742 |
| No | 188 | 59 | 1.00 | | 1.00 | |
| Antimicrobial and other medication | | | | | | |
| Metronidazole | | | | | | |
| Yes | 30 | 9 | 0.94 (0.42–2.10) | 0.878 | 1.03 (0.46–2.33) | 0.939 |
| No | 169 | 54 | 1.00 | | 1.00 | |
| Vancomycin | | | | | | |
| Yes | 3 | 2 | 2.13 (0.35–13.05) | 0.413 | 2.34 (0.37–13.73) | 0.384 |
| No | 195 | 61 | 1.00 | | 1.00 | |
| Amoxicillin + clavulanic acid | | | | | | |
| Yes | 13 | 2 | 0.47 (0.10–2.15) | 0.331 | 0.54 (0.12–2.48) | 0.427 |
| No | 187 | 61 | 1.00 | | 1.00 | |
| Piperacillin + tazobactam | | | | | | |
| Yes | 33 | 14 | 1.45 (0.72–2.92) | 0.303 | 1.31 (0.63–2.75) | 0.471 |
| No | 167 | 49 | 1.00 | | 1.00 | |
| Third-generation cephalosporins | | | | | | |
| Yes | 39 | 13 | 1.07 (0.53–2.17) | 0.844 | 1.18 (0.58–2.40) | 0.651 |
| No | 161 | 50 | 1.00 | | 1.00 | |
| Carbapenems | | | | | | |
| Yes | 34 | 12 | 1.45 (0.55–2.38) | 0.709 | 1.14 (0.53–2.42) | 0.743 |
| No | 166 | 51 | 1.00 | | 1.00 | |
| Fluoroquinolones | | | | | | |
| Yes | 39 | 9 | 0.69 (0.31–1.51) | 0.352 | 0.75 (0.34–1.66) | 0.476 |
| No | 161 | 54 | 1.00 | | 1.00 | |
| Azithromycin | | | | | | |
| Yes | 11 | 2 | 0.56 (0.12–2.61) | 0.463 | 0.59 (0.13–2.75) | 0.502 |
| No | 189 | 61 | 1.00 | | 1.00 | |
| Sulfamethoxazole + trimethoprim | | | | | | |
| Yes | 9 | 2 | 0.70 (0.15–3.31) | 0.648 | 0.73 (0.15–3.48) | 0.693 |
| No | 191 | 61 | 1.00 | | 1.00 | |
| Antiparasitic agents | | | | | | |
| Yes | 13 | 2 | 0.47 (0.10–2.13) | 0.325 | 0.49 (0.11–2.23) | 0.356 |
| No | 185 | 61 | 1.00 | | 1.00 | |
| Corticosteroids | | | | | | |
| Yes | 50 | 16 | 1.00 (0.52–1.92) | 0.998 | 1.13 (0.58–2.19) | 0.724 |
| No | 147 | 47 | 1.00 | | 1.00 | |
| Statins | | | | | | |
| Yes | 15 | 5 | 1.05 (0.36–3.00) | 0.933 | 1.01 (0.32–3.22) | 0.989 |
| No | 182 | 58 | 1.00 | | 1.00 | |
| Other medications | | | | | | |
| Yes | 97 | 34 | 1.23 (0.70–2.18) | 0.47 | 1.14 (0.63–2.05) | 0.661 |
| No | 102 | 29 | 1.00 | | 1.00 | |
| 1 week prior to stool collection | | | | | | |
| Antimicrobial consumption | | | | | | |
| Yes | 56 | 25 | 1.61 (0.88–2.96) | 0.121 | 1.80 (0.97–3.36) | 0.064 |

**TABLE 1** Univariate logistic regression analysis of socio-demographic, clinical, and other factors associated with *C. difficile* carriage[a,b] (Continued)

| | Number of cases | | COR (95% CI) | P value | COR (95% CI) | P value |
|---|---|---|---|---|---|---|
| | Without *C. difficile* (N = 200) | With *C. difficile* (N = 63) | (263 patients) | | (251 patients) | |
| No | 123 | 34 | 1.00 | | 1.00 | |
| Corticosteroids | | | | | | |
| Yes | 24 | 9 | 1.25 (0.55–2.86) | 0.597 | 1.40 (0.60–3.22) | 0.436 |
| No | 170 | 51 | 1.00 | | 1.00 | |
| Proton pump inhibitors | | | | | | |
| Yes | 75 | 31 | 1.49 (0.84–2.63) | 0.176 | 1.73 (0.96–3.14) | 0.071 |
| No | 115 | 32 | 1.00 | | 1.00 | |
| Statins | | | | | | |
| Yes | 7 | 3 | 1.40 (0.35–5.59) | 0.635 | 1.46 (0.36–5.83) | 0.596 |
| No | 186 | 57 | 1.00 | | 1.00 | |
| Other medications | | | | | | |
| Yes | 74 | 27 | 1.18 (0.66–2.12) | 0.578 | 1.24 (0.68–2.28) | 0.480 |
| No | 110 | 34 | 1.00 | | 1.00 | |
| 4 weeks prior to stool collection | | | | | | |
| Antimicrobial consumption | | | | | | |
| Yes | 6 | 5 | 2.76 (0.81–9.42) | 0.105 | 2.89 (0.84–9.89) | 0.091 |
| No | 169 | 51 | 1.00 | | 1.00 | |
| Corticosteroids | | | | | | |
| Yes | 9 | 2 | 0.67 (0.14–3.19) | 0.615 | 0.70 (0.15–3.36) | 0.659 |
| No | 178 | 59 | 1.00 | | 1.00 | |
| Proton pump inhibitors | | | | | | |
| Yes | 22 | 5 | 0.68 (0.25–1.89) | 0.461 | 0.78 (0.28–2.20) | 0.644 |
| No | 165 | 55 | 1.00 | | 1.00 | |
| Other medications | | | | | | |
| Yes | 49 | 19 | 1.20 (0.64–2.26) | 0.572 | 1.10 (0.56–2.15) | 0.778 |
| No | 130 | 42 | 1.00 | | 1.00 | |

[a]COR, crude odds ratio; COPD, chronic obstructive pulmonary disease; OPD, outpatient department; GI, gastrointestinal.
[b]The bolded values indicate significance with a p-value of <0.05.

## DISCUSSION

To the best of our knowledge, this is the first epidemiological study of *C. difficile* carriage in hospitalized adults in Cambodia. Since the study did not screen for cases of CDI due to the lack of diagnostic facilities and the lack of awareness of CDI in the country, it is reasonable to assume that most of the *C. difficile* carriage identified in this study was not in conjunction with clinical signs and symptoms of CDI and probably reflected

**TABLE 2** Multivariable logistic regression model of factors associated with *C. difficile* carriage (N = 263)[b]

| Feature | Number of cases | | AOR[a] (95% CI) (263 patients) | P value |
|---|---|---|---|---|
| | Without *C. difficile* (N = 200) | With *C. difficile* (N = 63) | | |
| Bowel function disorders | | | | |
| Yes | 2 | 3 | 9.91 (0.78–125.79) | 0.077 |
| No | 198 | 60 | 1.00 | |
| Diabetes | | | | |
| Yes | 65 | 30 | **2.48 (1.16–5.30)** | **0.019** |
| No | 135 | 33 | 1.00 | |
| Hospitalization >24 h within the last 3 months | | | | |
| Yes | 46 | 25 | **3.89 (1.79–8.43)** | **0.001** |
| No | 148 | 37 | 1.00 | |

[a]AOR, adjusted odds ratio.
[b]The bolded values indicate significance with a p-value of <0.05.

TABLE 3 Features of diarrheic patients from whom *C. difficile* was isolated

| Characteristic | Patient 1 | Patient 2 | Patient 3 | Patient 4 | Patient 5 |
|---|---|---|---|---|---|
| *C. difficile* strain | | | | | |
| Ribotype | Unique singleton | QX 011 | Unique singleton | 010 | QX 716 |
| Toxin profile | A−B−CDT− | A−B−CDT− | A+B+CDT− | A−B−CDT− | A−B−CDT− |
| Socio-demographic feature | | | | | |
| Sex | F | F | F | F | F |
| Age (years) | 66 | 54 | 26 | 68 | 50 |
| Residency | Province | Province | Phnom Penh | n/a | Province |
| Profession/career | Unemployed | Farmer | Petrol station attendant | Unemployed | Unemployed |
| Hospitalization and clinical feature | | | | | |
| Length of stay (days) | 7 | 6 | 6 | 7 | 5 |
| Diarrhea | ☑ | ☑ | ☑ | ☑ | ☑ |
| Abdominal pain | ☑ | ☑ | ☑ | ☑ | ☑ |
| Fever | ☑ | ☑ | ☑ | ☑ | |
| Diabetes | ☑ | | | ☑ | ☑ |
| Renal diseases | ☑ | | ☑ | ☑ | ☑ |
| Non-surgical gastro-intestinal procedure | | ☑ | | ☑ | |
| Medication | | | | | |
| Antiparasitic agents | ☑ | | ☑ | | |
| Corticosteroids | ☑ | | ☑ | ☑ | |
| Proton pump inhibitors | ☑ | ☑ | ☑ | ☑ | ☑ |
| Current other medications | ☑ | ☑ | | ☑ | |
| Antimicrobial consumption of the week before | ☑ | ☑ | ☑ | | |
| PPIs of the week before | ☑ | ☑ | ☑ | | |
| Other medications of the week before | ☑ | | ☑ | ☑ | |

colonization. The overall prevalence of *C. difficile* carriage in the current study was similar to that in patients in Thailand (24% Cambodia vs 23.7% Thailand); however, this was noticeably (though not significantly) lower when the comparison was made between diarrheic patients in the two countries (15.2% Cambodia vs 23.7% Thailand, $P = 0.262$) (23). The prevalence of *C. difficile* in diarrheic patients in Cambodia was identical to that in Vietnam (15.2% vs 15.1%, respectively) (20), possibly due to similarities in risk factors and healthcare systems in Cambodia and Vietnam, although this requires further investigation. A recent publication on environmental contamination with *C. difficile* in Vietnam found *C. difficile* in food and pig farms and in the hospital environment including floors, beds, toilets, devices, gardens, and playgrounds (29). These findings provide more insights into potential sources of *C. difficile* in hospital and community settings in the Cambodian context which appears similar to Vietnam.

## Risk factors

Well-known risk factors for *C. difficile* carriage, including antimicrobial consumption, PPI use, aging, and comorbidities including diabetes and malignancy, have been discussed extensively previously (11–13, 15). In the current study, diabetes (OR = 2.56; 95% CI: 1.14–5.74) and hospitalization >24 h within the last 3 months (OR = 4.86; 95% CI: 2.06–11.48) were significantly associated with *C. difficile* carriage. Diabetes prevalence in Cambodia ranged from 5% to 11% in 2004, and an increase is expected in future years (30). The decrease in bacterial diversity in the gut of diabetic patients is a risk factor for acquiring *C. difficile* (31). The hospital environment is a well-known reservoir of *C. difficile* spores (29, 32, 33). Hospital beds, light switches, floors, toilets, sinks, and other objects can be contaminated with *C. difficile* if a *C. difficile*-positive patient occupies the room. A hospital bed could remain contaminated with spores for up to 90 days, posing a high risk of colonization and disease for the next patient (33).

The high proportion of positive cases in patients with nephrotic syndrome and OVB suggests the possibility that these two conditions are important in *C. difficile* acquisition

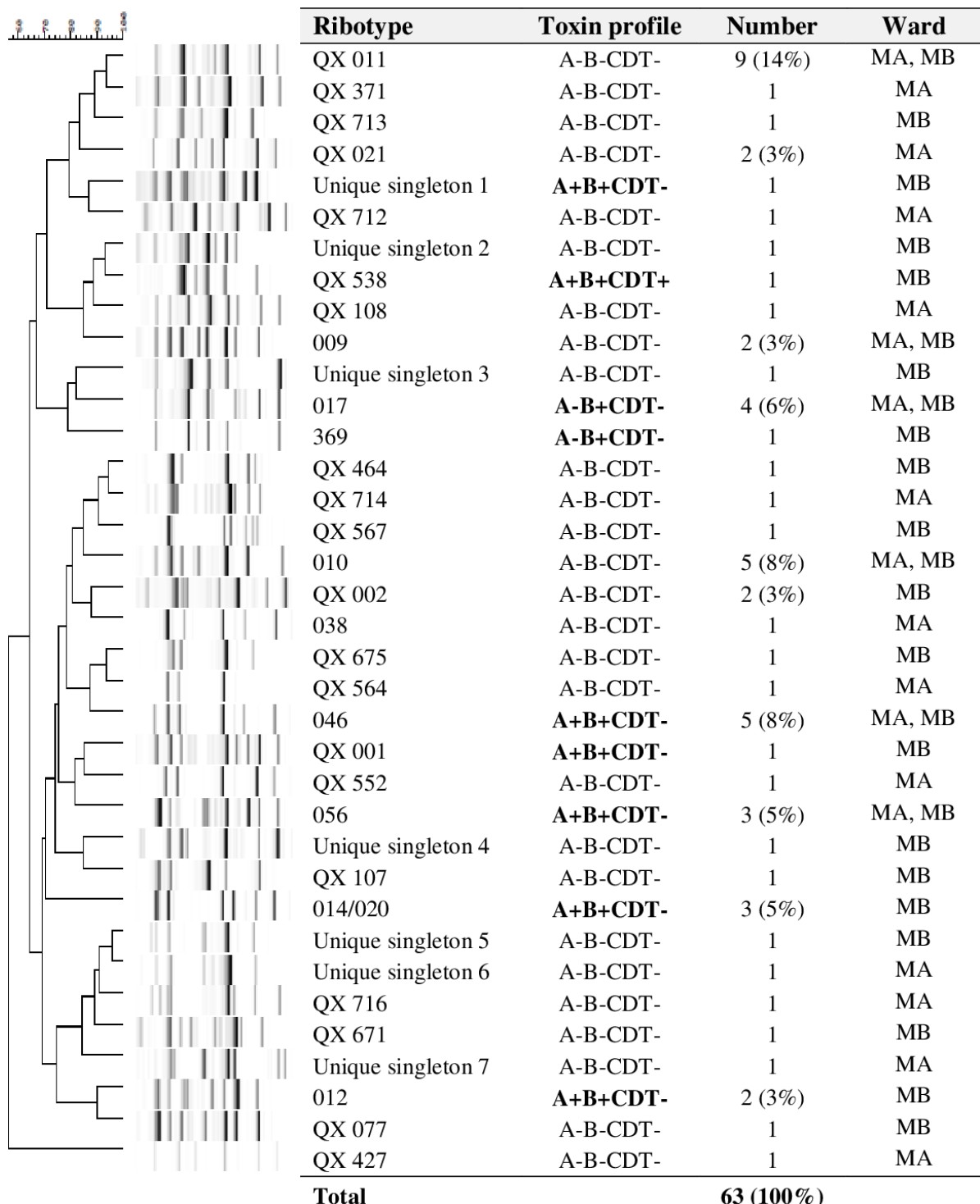

| Ribotype | Toxin profile | Number | Ward |
|---|---|---|---|
| QX 011 | A-B-CDT- | 9 (14%) | MA, MB |
| QX 371 | A-B-CDT- | 1 | MA |
| QX 713 | A-B-CDT- | 1 | MB |
| QX 021 | A-B-CDT- | 2 (3%) | MA |
| Unique singleton 1 | **A+B+CDT-** | 1 | MB |
| QX 712 | A-B-CDT- | 1 | MA |
| Unique singleton 2 | A-B-CDT- | 1 | MB |
| QX 538 | **A+B+CDT+** | 1 | MB |
| QX 108 | A-B-CDT- | 1 | MA |
| 009 | A-B-CDT- | 2 (3%) | MA, MB |
| Unique singleton 3 | A-B-CDT- | 1 | MB |
| 017 | **A-B+CDT-** | 4 (6%) | MA, MB |
| 369 | **A-B+CDT-** | 1 | MB |
| QX 464 | A-B-CDT- | 1 | MB |
| QX 714 | A-B-CDT- | 1 | MA |
| QX 567 | A-B-CDT- | 1 | MB |
| 010 | A-B-CDT- | 5 (8%) | MA, MB |
| QX 002 | A-B-CDT- | 2 (3%) | MB |
| 038 | A-B-CDT- | 1 | MA |
| QX 675 | A-B-CDT- | 1 | MB |
| QX 564 | A-B-CDT- | 1 | MA |
| 046 | **A+B+CDT-** | 5 (8%) | MA, MB |
| QX 001 | **A+B+CDT-** | 1 | MB |
| QX 552 | A-B-CDT- | 1 | MA |
| 056 | **A+B+CDT-** | 3 (5%) | MA, MB |
| Unique singleton 4 | A-B-CDT- | 1 | MB |
| QX 107 | A-B-CDT- | 1 | MB |
| 014/020 | **A+B+CDT-** | 3 (5%) | MB |
| Unique singleton 5 | A-B-CDT- | 1 | MB |
| Unique singleton 6 | A-B-CDT- | 1 | MA |
| QX 716 | A-B-CDT- | 1 | MA |
| QX 671 | A-B-CDT- | 1 | MB |
| Unique singleton 7 | A-B-CDT- | 1 | MA |
| 012 | **A+B+CDT-** | 2 (3%) | MB |
| QX 077 | A-B-CDT- | 1 | MB |
| QX 427 | A-B-CDT- | 1 | MA |
| **Total** | | **63 (100%)** | |

**FIG 3** *C. difficile* ribotyping banding patterns and toxin profiles, by frequency and ward.

(Table S1). Nephrotic syndrome could be an initial step in renal disease development (34). Patients with renal diseases, mostly chronic renal failure, are at risk of *C. difficile* acquisition and CDI development due to impaired immune defense, frequent antimicrobial consumption for bacterial infections, and other medications (34). Though renal diseases were not associated with *C. difficile* carriage in the current study, several studies in Asia have found renal diseases to be a risk factor for CDI (11, 20). Though there is not

enough evidence to indicate transmission of *C. difficile* via endoscopy (35), a recent study found that patients with OVB were at higher risk of CDI compared to non-OVB patients (36). While it can be hypothesized that good cleaning procedures for endoscopic devices at each hospital could mitigate this risk, cleaning of endoscopes remains a topical issue (35). Further investigation into the association between nephrotic syndrome, renal diseases, OVB, and the acquisition of *C. difficile* or CDI in patients with these conditions in Cambodia should be considered.

## Molecular prevalence

The toxin A-negative *C. difficile* RT 017 strain is endemic in Asia as seen in several previous studies in Asia (11, 16, 19, 37). In the current study, *C. difficile* RT 046 (8%, 5/63) was slightly more prevalent than RT 017 (6%, 4/63) (Fig. 3); however, the difference was not statistically significant ($P = 0.729$). *C. difficile* RT 046 was reported recently from an outbreak in a Swedish hospital but is a less common RT in many hospitals in Europe and in multiple studies in Asia (13, 20, 21, 37, 38). *C. difficile* RT 046 was reported to spread effectively in the hospital environment, possibly due to its multidrug resistance, and contributed to more severe disease compared to other RTs with high mortality in outbreaks (38–40). Interspecies *C. difficile* RT 046 transmission between humans and pigs was also reported, suggesting community sources of this strain (41). The predominance of RT 046 and RT 017 in this study requires further investigations in the hospital environment in Cambodia, and multiple factors including infection prevention and control (IPC) measures, antimicrobial stewardship, and *C. difficile* sources should be taken into consideration. CDI diagnostic procedures are not yet well established in many low-income countries in Asia, and thus IPC measures targeting *C. difficile* are not in place, so silent circulation of *C. difficile* in the hospital is not surprising.

The findings of overlapping RTs including *C. difficile* RTs 012, 014/020, 056, 009, and 010 from hospitalized patients and animals/food/environment suggest the possibility of transmission from community to hospital settings or vice versa (29, 32, 42–44). The prevalence of *C. difficile* in the 12 patients hospitalized <48 h was not significantly higher than the prevalence in 251 patients ($P = 0.141$). The five patients hospitalized <48 h who had *C. difficile* may have been asymptomatic carriers and more likely acquired *C. difficile* in the community. Transmission from community to hospital settings is quite possible in this context.

The prevalence of binary toxin-positive strains of *C. difficile* was low, and an isolate of *C. difficile* QX538 was the only one among 63 isolates (1.6%) in this study. Such strains are rarely reported in Asia and are most prevalent in North America and Europe (9, 10, 45). A 10-year study in South Korea showed a similar prevalence to Cambodia at 1.8% (58/3,278) (46). All these Korean binary toxin-positive isolates belonged to evolutionary clades 2, 3, and 5 except two isolates that were unknown. Another study in Thailand collected 321 strains and found three *C. difficile* isolates with the toxin profile A+B+CDT+ (0.93%), one of which was RT 078 in clade 5 and the other two were of an unknown RT (37). Pathogenicity of binary toxin-positive epidemic *C. difficile* RT 027 possibly depends on increased production of toxins A and B related to an 18 bp deletion in *tcdC* and production of binary toxin (47, 48). Binary toxin-positive strains of *C. difficile* are known to also cause more severe disease compared to binary toxin-negative strains (49); however, *C. difficile* QX538 in the current study was isolated from a non-diarrheic patient. Only one of the toxigenic strain-carrying patients reported diarrhea, and it can be speculated that the asymptomatic status of this patient is attributable to protective antibodies or prophylactic potential from previous non-toxigenic *C. difficile* strain carriage or other local protective factors to be determined, including the gut microbiota composition (16, 17, 50–52).

Based on genomic analysis by multi-locus sequence type (MLST), except *C. difficile* RT 017 (ST37) a clade 4 RT, other RTs 046 (ST35), 056 (ST58), 014/020 (ST2), 012 (ST54), and 038 (ST48) were classified in evolutionary clade 1 (53, 54). Based on the comparison with reference strains in the laboratory, the long list of diverse strains with QX prefixes

(Fig. 3) is more likely to fall into clade 4. With the plasticity and ultralow conservation of the *C. difficile* genome, the most recent occurrence of multiple exchanges, loss, and acquisitions of the pathogenicity locus (PaLoc) in the *C. difficile* genome was estimated at ~50 years ago in clade 1 (55). *C. difficile* RT 017 was estimated to have acquired the PaLoc ~500 years ago (55, 56). Since *C. difficile* was shown to have common ancestors from between 0.46 and 75 million years ago (57), it can be hypothesized that the QX strains from the current study are either unidentified diverging strains or local strains in Asia, likely in clade 4 (58), as mentioned above. Further genotypic identification of strains by whole genome sequencing is pivotal in determining transmission routes and predicting possible future outbreaks. With a genome of high plasticity, *C. difficile* is likely to be able to acquire and disseminate resistance determinants intra-species and inter-species (54, 59, 60). Thus, the high prevalence of asymptomatic carriers of both toxigenic and non-toxigenic *C. difficile* strains, and poorly controlled antimicrobial usage in Cambodia, could lead to a possible rise in AMR in the future.

There were some limitations in this study. It was not possible to confirm if patients had asymptomatic CDI or were at an early stage of the infection cycle because CDI diagnostics were not implemented. This was a cross-sectional study with no follow-up of patients; thus, later development of the disease could not be ascertained. Also, the absence of antimicrobial susceptibility results in this study limits the discussion on treatment options for Cambodia.

In conclusion, the high prevalence of asymptomatic CDI in hospitalized adults in Cambodia indicates the presence of *C. difficile* that needs to be identified and managed. The scarcity of epidemiological data on *C. difficile* in Cambodia contributes to the misdiagnosis of true CDI cases. Risk factors including diabetes and hospitalization >24 h within the last 3 months were associated with *C. difficile* asymptomatic carriage; however, risk factors for disease require further investigation. The five most predominant toxigenic *C. difficile* strains were RTs 046, 017, 056, 014/020, and 012, while non-toxigenic strains included RTs QX011, 010, 009, QX021, and QX002. The presence of *C. difficile* in patients residing in almost every province of Cambodia indicates the likelihood of *C. difficile* endemicity in Cambodia. The impacts of toxigenic and non-toxigenic *C. difficile* carriage are of interest in CDI management, IPC implementation, and AMR risk assessment and management in Cambodia. Genotypic identification of local *C. difficile* strains in Cambodia is necessary for a better understanding of the epidemiology of CDI and the importance of *C. difficile*.

## MATERIALS AND METHODS

### Study setting

The study was conducted at CH, a 1,050-bed referral hospital located in the capital of Cambodia, Phnom Penh.

### Sample collection and transport

Inpatients at CH aged ≥18 years were recruited from MA and MB wards between June and September 2022 with informed consent. The exclusion criteria were a lack of informed consent and a predicted hospitalization of <48 h. Participants were requested to provide one stool sample during their stay at the hospital. Stool samples were collected in a conventional stool pot by the participants. Upon arrival at the laboratory, the consistency of each stool sample was noted, based on the Bristol Stool Form Scale: hard stool (types 1 and 2), normal stool (types 3, 4, and 5), and loose stool (types 6 and 7) (61). The sample was then stored at 2–8°C before a Transwab (Medical Wire and Equipment Co. Ltd., England) was used to collect feces for long-term storage at ambient temperature at CH and subsequent transport to Australia. Transportation at ambient temperature from Cambodia to the reference laboratory at the Queen Elizabeth II Medical Centre in Nedlands, Western Australia, took 7–10 days and was undertaken in

October 2023. This process of transportation was used successfully in several previous studies of *C. difficile* in other Asian countries (16, 19, 20).

## Detection of *C. difficile*

ChromID *C. difficile* agar (bioMérieux, Marcy l'Etoile, France) was used for direct culture of stool samples with incubation in an A35 anaerobic chamber (Don Whitley Scientific, Ltd., Shipley, West Yorkshire, United Kingdom) at 35°C for 48 h, with an atmosphere of 80% $N_2$, 10% $CO_2$, 10% $H_2$, and 75% relative humidity. Indirect culture was performed by enrichment of specimens in supplemented Robertson's cooked meat medium (PathWest Laboratory Medicine Excel Media, Mount Claremont, Western Australia) containing 250 mg/L cycloserine, 8 mg/L cefoxitin, and 5 mg/L gentamicin, incubated in aerobic conditions at 35°C for 4–7 days. Alcohol shock with equal volumes of cooked meat medium broth and absolute alcohol (1 mL) for 1 h was done, and the mixture then was plated on cycloserine cefoxitin fructose agar (CCFA) (PathWest Excel Media). Putative *C. difficile* colonies on ChromID *C. difficile* agar and CCFA were sub-cultured onto horse blood agar (BA) plates, and the putative identity was confirmed by the typical colonial morphology on BA, horse dung odor, and chartreuse fluorescence under UV light (62).

## Toxin gene detection and ribotyping

Heat DNA extraction was performed with 5% Chelex-100 (Sigma-Aldrich, Castle Hill, NSW, Australia) from pure 48–72 h cultures on BA. The toxin gene *tcdA* was detected in a duplex PCR using NK2/NK3 primers for the non-repeating sequence *tcdA1* (63) and novel BEtcdA1/BEtcdA2 primers for repeating sequence *tcdA3* (64). The toxin B gene (*tcdB*) and binary toxin genes (*cdtA* and *cdtB*) were detected in monoplex PCRs using NK104/NK105, cdtApos/cdtArev, and cdtBpos/cdtBrev primers, respectively (65, 66). PCR ribotyping was performed as described by Stubbs et al. with some modifications (67). PCR products were cleaned with a MinElute PCR purification kit (Qiagen, Venlo, Limburg, The Netherlands). PCR products for toxin gene detection and ribotyping were analyzed on a QIAxcel capillary electrophoresis platform (Qiagen, Venlo, Limburg, The Netherlands). Ribotyping banding profiles of each isolate were compared to a collection of reference strains in the laboratory using BioNumerics software package version 7.6.3 (Applied Maths, Saint-Martens-Latem, Belgium). A prefix of "QX" was assigned to strains unmatched or unknown in the strain collection.

## Demographic and clinical information

Comprehensive demographic and clinical data were collected from each participant, including the region of residence, presence of siblings under 1 year of age, occupation, living close to livestock, and previous hospitalizations and LOS. A detailed medical history was recorded at the time of admission, including the diagnosis, symptoms such as diarrhea, abdominal pain, and fever, as well as underlying clinical conditions including diabetes, chronic obstructive pulmonary disease, heart failure, stroke, hematological malignancy, cancer, and medical interventions such as surgery and non-surgical gastrointestinal procedures. Additionally, the frequency of OPD visits (at least 1/week in the last 4 weeks) and medication use including antimicrobials, antiparasitic agents, chemotherapy, corticosteroids, immunosuppressants, PPIs, $H_2$ receptor antagonists, probiotics, statins, gastrostomy feeding, hemodialysis, and other medications were collected. Treatment data were categorized based on the timeframe relative to stool collection, as either currently ongoing, within 1 week prior, or within 4 weeks prior.

## Statistical analysis

Descriptive statistics were used to summarize demographic and clinical variables by calculating percentages and frequencies. Univariate logistic regression analyses were

initially performed to establish risk factors for carriage of *C. difficile*, and crude odds ratios with 95% CIs were reported. Variables with a *P* value <0.2 in the univariate analysis were selected for the final model and assessed for multicollinearity using variance inflation factors (VIFs), with variables having a VIF > 5 excluded. Multivariable logistic regression models with forward stepwise methods were then used to assess the association between independent and dependent variables. Adjusted odds ratios with 95% CI were calculated to measure the strength of these associations. Variables with a *P* value <0.05 in the multivariable analysis were considered statistically significant for association with *C. difficile* carriage. All analyses were conducted using SPSS version 26.0.0.0.

## ACKNOWLEDGMENTS

We are grateful to the laboratory team at Calmette Hospital for assisting with sample collection and storage before sending samples to Western Australia.

T.V.R. has received a grant from Roche Diagnostics outside the present work.

## AUTHOR AFFILIATIONS

[1]School of Population Health, Faculty of Health Sciences, Curtin University, Bentley, Western Australia, Australia
[2]Calmette Hospital, Phnom Penh, Cambodia
[3]School of Medical and Health Sciences, Edith Cowan University, Joondalup, Western Australia, Australia
[4]School of Biomedical Sciences, The University of Western Australia, Perth, Western Australia, Australia
[5]The Kids Research Institute Australia, Perth, Western Australia, Australia
[6]The University of Health Sciences, Phnom Penh, Cambodia
[7]PathWest Laboratory Medicine, Perth, Western Australia, Australia

## PRESENT ADDRESS

Deirdre A. Collins, AMRIC, Health Service Executive, Dublin, Ireland
Archie C. A. Clements, Queen's University Belfast, Belfast, United Kingdom

## AUTHOR ORCIDs

Lengsea Eng http://orcid.org/0000-0002-7631-8678
Thomas V. Riley http://orcid.org/0000-0002-1351-3740

## AUTHOR CONTRIBUTIONS

Lengsea Eng, Conceptualization, Data curation, Formal analysis, Investigation, Methodology, Project administration, Resources, Validation, Visualization, Writing – original draft, Writing – review and editing | Deirdre A. Collins, Conceptualization, Methodology, Supervision, Writing – review and editing | Kefyalew Addis Alene, Formal analysis, Methodology, Supervision, Writing – review and editing | Sotharith Bory, Project administration | Youdaline Theng, Project administration | Pisey Vann, Project administration | Sreyhuoch Meng, Project administration | Setha Limsreng, Project administration | Archie C. A. Clements, Conceptualization, Formal analysis, Methodology, Supervision, Writing – review and editing | Thomas V. Riley, Conceptualization, Formal analysis, Investigation, Methodology, Project administration, Resources, Supervision, Validation, Visualization, Writing – review and editing

## ETHICS APPROVAL

Ethical approval for the project was provided by the National Ethics Committee for Health Research in Cambodia (248NECHR) and by the Human Research Ethics Office of Curtin University in Perth, Western Australia (HRE2022-0027).

## ADDITIONAL FILES

The following material is available online.

### Supplemental Material

**Supplemental material (Spectrum02747-24-s0001.pdf).** Tables S1 and S2.

### Open Peer Review

**PEER REVIEW HISTORY (review-history.pdf).** An accounting of the reviewer comments and feedback.

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
