## [Reviewer comments · Microbiology Spectrum]

Microbiology Spectrum

***Clostridioides (Clostridium) difficile* infection in hospitalised adult patients in Cambodia.**

Lengsea Eng, Deirdre Collins, Kefyalew Alene, Sotharith Bory, Youdaline Theng, Pisey Vann, Sreyhuoch Meng, Setha Limsreng, Archie Clements, and Thomas Riley

Corresponding Author(s): Lengsea Eng, Curtin University

Review Timeline:

Submission Date:	October 30, 2024
Editorial Decision:	December 4, 2024
Revision Received:	December 16, 2024
Accepted:	December 17, 2024

Editor: Karen Carroll

Reviewer(s): Disclosure of reviewer identity is with reference to reviewer comments included in decision letter(s). The following individuals involved in review of your submission have agreed to reveal their identity: Dazhi Jin (Reviewer #1)

Transaction Report:

DOI: <https://doi.org/10.1128/spectrum.02747-24>

Re: Spectrum02747-24 (*Clostridioides (Clostridium) difficile* infection in hospitalised adult patients in Cambodia.)

Dear Ms. Lengsea Eng:

Thank you for the privilege of reviewing your work. Your manuscript has been reviewed by two experts in the field. I concur with the suggestions by both reviewers to include information on antimicrobial resistance if it is available to complete your otherwise comprehensive epidemiological study. If that information is not available, I suggest shortening the existing discussion of AMR and noting the lack of AMR data as a potential limitation.

Below you will find instructions from the Spectrum editorial office, and the reviewer comments.

Revision Guidelines

Sincerely,
Karen Carroll
Editor
Microbiology Spectrum

Reviewer #1 (Comments for the Author):

The study reported by Lengsea Eng et al described *C. difficile* carriage in hospitalized adult in Cambodia including genotypes

and risk factors. However, the current version should be substantially improved as below.

Major comments:

1. Most of patients involved in this study did not present diarrhea. So, this is a study focusing on *C. difficile* carriage for genotyping and risk factors. The title and the related statement should be revised.
2. The authors should provide exact percentage of each dominant genotypes in Abstract and Results. As Fig 3, each of dominant genotypes was extremely few. I think that statistical analysis was not meaningful.
3. The data on antibiotic resistance testing were very important to disclose molecular epidemiology of CDI. But, no data were presented.
4. More data should be introduced in Asia-Pacific regions including Middle, East, and South Asia.
5. A total of 12 patients hospitalized for less than 48 h at the time of sample collection. Why was a predicted hospitalisation of less than 48 h chosen as one of the exclusion criteria?
6. In lines 268-269, the BSFS was used to note stool sharps, but no results were associated with it.
7. In lines 273-274, I am very confused this statement. The data on several studies used the data in this study, or not?
8. I think that it is reasonable that no relationships between antimicrobial consumption and *C. difficile* carriage. Why were demographic factors chosen for investigation of risk factors in patients with *C. difficile* carriage? Were there some correlations between them?

Minor comments:

1. In line 68, a reference is needed.
2. In line 73, The Philippines, that is Philippines.
3. In line 109, the full name of LOS should be provided.
4. In line 231, A study on *C. difficile*.....
5. In Table, the number of a total of cases with or without *C. difficile* should be presented.
6. The solution figures should be improved.

Reviewer #3 (Comments for the Author):

Dear authors,

Many thanks for the opportunity to review this interesting article.

It is important to assess the impact of *C. difficile* also in countries where it is believed to have a rather low impact.

In the result section, you state something about different regions of Cambodia. It would be beneficial to see the catchment area of the hospital, e.g. in terms of a map.

Why are resistance data not included? This would be highly beneficial.

Minor: In the Abstract I would just state e.g. Tertiary Care University center instead of the name.

line 61: I am not a native speaker but is there a grammar error present?

line 68: please give a reference. Is this truly the case?

With the best regards

Response to Reviewers

Reviewer #1 (Comments for the Author):

The study reported by Lengsea Eng et al described *C. difficile* carriage in hospitalized adult in Cambodia including genotypes and risk factors. However, the current version should be substantially improved as below.

Major comments:

1. Most of patients involved in this study did not present diarrhea. So, this is a study focusing on *C. difficile* carriage for genotyping and risk factors. The title and the related statement should be revised.

Infection does not mean disease! Individuals can get infected by C. difficile and subsequently remain colonised or develop disease, based on host susceptibility. Since this is the first study of C. difficile infection in Cambodia, we aimed to detect infection with C. difficile in the study population. Given there were no diagnostic tools available in the institution, it was not possible to confirm disease.

Thus, there is no need to change the title.

2. The authors should provide exact percentage of each dominant genotypes in Abstract and Results. As Fig 3, each of dominant genotypes was extremely few. I think that statistical analysis was not meaningful.

The percentages of each dominant genotype are now provided in the Abstract and Results. Although the numbers are low, it is worth seeing the significant differences among the genotypes.

3. The data on antibiotic resistance testing were very important to disclose molecular epidemiology of CDI. But, no data were presented.

See the response to a similar query from Dr Carroll, the editor (below).

4. More data should be introduced in Asia-Pacific regions including Middle, East, and South Asia.

Many studies from Asia are already referred to in the manuscript – these are mainly from East and Southeast Asia although there is one from China. For completeness, we have added a couple of references from Japan and South Korea ie North Asia.

5. A total of 12 patients hospitalized for less than 48 h at the time of sample collection. Why was a predicted hospitalisation of less than 48 h chosen as one of the exclusion criteria?

*Some samples were provided by the recruited patients earlier than what was recommended ie ≥ 48 h. With the high prevalence of *C. difficile* (41.7%, 5/12) in patients hospitalised < 48 h, these patients were included because this provided an additional interesting finding ie many patients were likely to be bringing *C. difficile* into the hospital. Thus, it was included in the Results.*

6. In lines 268-269, the BSFS was used to note stool sharps (?shapes), but no results were associated with it.

This description of stool was used a few times in the manuscript eg Table 1 where the presence of C. difficile in loose stool (COR = 0.74, 95%CI: 0.21 – 2.58) was compared to normal and hard stool. To avoid confusion, extra information was added about BSFS.

7. In lines 273-274, I am very confused this statement. The data on several studies used the data in this study, or not?

No, those are studies conducted earlier in other Asian countries using the same transport method. The statement is to show the suitability of the current study method compared to previous studies.

8. I think that it is reasonable that no relationships between antimicrobial consumption and C. difficile carriage. Why were demographic factors chosen for investigation of risk factors in patients with C. difficile carriage? Were there some correlations between them?

*Because some of those demographic factors are known risk factors eg age! All the results or correlations are shown in **Table 1**.*

Minor comments:

1. In line 68, a reference is needed.

Three studies are now cited.

2. In line 73, The Phillipies, that is Phillipies.

The name of the country is “The Philippines”

3. In line 109, the full name of LOS should be provided.

It is provided now.

4. In line 231, A study on C. difficile.....

Although there is nothing wrong with this statement it is no longer in the manuscript. This section has been modified, and reduced as suggested by Dr Carroll, due to the lack of antimicrobial susceptibility results in this study.

5. In Table, the number of a total of cases with or without C. difficile should be presented.

The numbers are now provided.

6. The solution (?resolution) figures should be improved.

All figure files were converted into TIFF files for high-resolution purposes.

Reviewer #3 (Comments for the Author):

Dear authors,

Many thanks for the opportunity to review this interesting article.

It is important to assess the impact of C. difficile also in countries where it is believed to have a rather low impact.

In the result section, you state something about different regions of Cambodia. It would be beneficial to see the catchment area of the hospital, e.g. in terms of a map.

There is a map as part of Figure 2.

Why are resistance data not included? This would be highly beneficial.

See the response to Dr Carroll, the editor (below).

Minor: In the Abstract I would just state e.g. Tertiary Care University center instead of the name.

This is the first study of CDI in Cambodia; thus, mentioning the name of the institution is appropriate for the audience. In addition, several authors work in that hospital.

line 61: I am not a native speaker but is there a grammar error present?

The sentence is grammar error-free.

line 68: please give a reference. Is this truly the case?

*See the response to **Reviewer #1**.*

Response to the editor

The antimicrobial susceptibility testing (AST) results were not available when this manuscript was prepared. In addition, we are conducting several studies in different populations in Cambodia, and it is likely that the AST results will be the basis for a separate manuscript on the total collection of strains from those studies. Thus, we have followed your suggestion and reduced the discussion around AMR a little.

Re: Spectrum02747-24R1 (*Clostridioides (Clostridium) difficile* infection in hospitalised adult patients in Cambodia.)

Dear Ms. Lengsea Eng:

Your manuscript has been accepted, and I am forwarding it to the ASM production staff for publication. Your paper will first be checked to make sure all elements meet the technical requirements. ASM staff will contact you if anything needs to be revised before copyediting and production can begin. Otherwise, you will be notified when your proofs are ready to be viewed.

Sincerely,
Karen Carroll
Editor
Microbiology Spectrum